# Conventional and Deep-Learning-Based Image Reconstructions of Undersampled K-Space Data of the Lumbar Spine Using Compressed Sensing in MRI: A Comparative Study on 20 Subjects

**DOI:** 10.3390/diagnostics13030418

**Published:** 2023-01-23

**Authors:** Philipp Fervers, Charlotte Zaeske, Philip Rauen, Andra-Iza Iuga, Jonathan Kottlors, Thorsten Persigehl, Kristina Sonnabend, Kilian Weiss, Grischa Bratke

**Affiliations:** 1Institute for Diagnostic and Interventional Radiology, Faculty of Medicine and University Hospital Cologne, 50937 Cologne, Germany; 2Philips GmbH Market DACH, 22335 Hamburg, Germany

**Keywords:** magnetic resonance imaging, artificial intelligence, image processing, computer-assisted

## Abstract

Compressed sensing accelerates magnetic resonance imaging (MRI) acquisition by undersampling of the k-space. Yet, excessive undersampling impairs image quality when using conventional reconstruction techniques. Deep-learning-based reconstruction methods might allow for stronger undersampling and thus faster MRI scans without loss of crucial image quality. We compared imaging approaches using parallel imaging (SENSE), a combination of parallel imaging and compressed sensing (COMPRESSED SENSE, CS), and a combination of CS and a deep-learning-based reconstruction (CS AI) on raw k-space data acquired at different undersampling factors. 3D T2-weighted images of the lumbar spine were obtained from 20 volunteers, including a 3D sequence (standard SENSE), as provided by the manufacturer, as well as accelerated 3D sequences (undersampling factors 4.5, 8, and 11) reconstructed with CS and CS AI. Subjective rating was performed using a 5-point Likert scale to evaluate anatomical structures and overall image impression. Objective rating was performed using apparent signal-to-noise and contrast-to-noise ratio (aSNR and aCNR) as well as root mean square error (RMSE) and structural-similarity index (SSIM). The CS AI 4.5 sequence was subjectively rated better than the standard in several categories and deep-learning-based reconstructions were subjectively rated better than conventional reconstructions in several categories for acceleration factors 8 and 11. In the objective rating, only aSNR of the bone showed a significant tendency towards better results of the deep-learning-based reconstructions. We conclude that CS in combination with deep-learning-based image reconstruction allows for stronger undersampling of k-space data without loss of image quality, and thus has potential for further scan time reduction.

## 1. Introduction

The growing and aging world population has an ever-increasing demand for magnetic resonance imaging (MRI) [1]. To meet this demand, recent technical developments aim to speed up MRI examinations, which increases imaging capacities [2,3,4]. One clinically established MRI acceleration technique is compressed sensing [5,6]. Compressed sensing achieves shorter MRI scanning times by undersampling data from the k-space during acquisition [5,6]. The k-space holds the MRI raw data before reconstruction into visually perceivable images [7]. Compressed sensing has already enabled a significant reduction of scan time in multiple settings, especially when working with 3D sequences [8,9,10,11,12]. Another established approach to achieving shorter MRI scanning times is parallel imaging, which exploits redundancy of spatial information to reduce the amount of sampled k-space data [13]. 

Recently, the combination of both compressed sensing and parallel imaging (Compressed SENSE, CS, Philips, the Netherlands), was shown to allow for higher acceleration of acquisition times while preserving the image quality [3,14,15]. Higher acceleration factors during image acquisition results in less sampling of k-space data, which, however, might impair image quality in cases of excessive undersampling. Artificial intelligence (AI)-based image reconstruction techniques were shown by Pezzotti et al. to be able to interpolate non-acquired data from acquired k-space data; therefore these techniques might allow for further improvement in image quality of accelerated data or further acceleration by compensating for image quality trade-offs [16,17,18]. The underlying algorithm architecture Adaptive-CS-Net comprises a convolutional neural network, which won the international fast MRI Facebook challenge in 2019 in the category “8x undersampling” [17,19]. This neural network was further developed by the manufacturer to include more than 740,000 undersampled MRI imagings of several contrasts, anatomies, and acceleration factors in the training data [20]. It was provided as a prototype by the MRI system manufacturer and its applicability on the lumbar spine was investigate in this study. The prototype AI-reconstruction algorithm was later adopted to a clinically approved software suite by the manufacturer without major changes [15].

MRI of the lumbar spine is considered one of the most frequently performed examinations and it therefore holds great potential for time saving. Important areas of use include, e.g., the work-up of lower back pain with neurological deficits or suspected underlying disease and the planning of surgical management in cases of radiculopathy and spinal stenosis as well as postoperative imaging [21,22]. Nevertheless, it is important to maintain diagnostic quality to provide certainty, especially in diagnosing cord or nerve-root compression syndromes. The purpose of this study was to compare deep-learning-based MRI reconstructions with conventional reconstructions of a 3D T2 sequence of the lumbar spine in terms of subjective and objective image quality. Specifically, we aimed to determine whether deep-learning-based reconstructions offer further opportunities to accelerate image acquisition by offsetting image quality trade-offs.

## 2. Materials and Methods

### 2.1. Study Population

This prospective single-center study was approved by the institutional review board and registered in the national Clinical Trials Register (DRKS00024156). Written informed consent was obtained from all participants included in the study. Inclusion criteria for the volunteers was age >18 years. Exclusion criteria were pregnancy, implanted MRI conditional or unsafe devices, previous surgery or known diseases of the spine, and lower back pain within the last 6 months. Imaging data were acquired from April 2021 to May 2021. 

### 2.2. MRI Protocol

MRI examination was performed in a whole-body 3T MRI system (Ingenia 3.0 T, Philips Healthcare) using the 12-channel in-built table coil array for signal reception. The position of the volunteers was supine, head-first on the table. 

First, a reference 3D T2 turbo spin echo (TSE) sequence, as provided by the manufacturer (including parallel imaging acceleration of 2.5), was acquired, referred to as standard SENSE. This was followed by 3D T2 TSE sequences using CS acceleration with acceleration factors of 4.5, 8, and 11, based on previous experiences of Bratke et al. [3]. As introduced above, CS exploits a combination of compressed sensing and parallel imaging for acceleration of MRI acquisition [23]. The three sets of undersampled k-space data were then reconstructed to visually perceivable images using 1) a conventional approach (CS) and 2) a novel AI-driven prototype (CS AI). The AI-prototype was based on the convolutional neural network “Adaptive-CS-Net”, which processes undersampled k-space data in an iterative, learning-based reconstruction scheme. In this way, the conventional wavelet transformation to process undersampled k-space data was replaced by a neural network. The algorithm uses a learning-based sparsifying approach with consistency checks to the raw k-space data in each block, to pursue maximum image authenticity [15,17]. In the following, the conventional and AI-driven reconstruction methods are referred to as CS and CS AI, respectively. Figure 1 illustrates the workflow of our study. Please see Appendix A for in-detail information about the MRI protocol. 

### 2.3. Image Analysis

Image analysis was performed using both an objective and subjective approach. In the objective approach, the analysis performed was region of interest (ROI)-based and pixel-based. 

### 2.4. Objective Image Analysis: ROI-Based

Since the iterative reconstruction of Compressed SENSE leads to an artificial noise reduction in the image that affects the background noise, classical ROI-based parameters such as the signal-to-noise ratio (SNR) and the contrast-to-noise ratio (CNR) are relevantly affected (depending on the weighting between data consistency and noise reduction during iterative reconstruction). The informative value of these parameters therefore appears to be restricted. Similar to the previously published studies by Bratke et al., we therefore decided to quantify potential differences by the apparent SNR (aSNR) and apparent CNR (aCNR) [3,14]. The aSNR was calculated by dividing the signal intensity by the standard deviation (SD) of the same ROI, while aCNR was calculated by subtracting the signal intensity of the different tissues divided by the SD [3,14]. The applied calculations to yield aCNR and aSNR are reported in a standard format as Equations (1) and (2). ROIs were drawn in the central slice of each sequence in the vertebral body of L1 with an area of 150 mm^2^, in the spinal cord in segment TH 11/12 with an area of 20 mm^2^ and in the cerebrospinal fluid (CSF) within the spinal canal in segment L3 with an area of 25 mm^2^. From these data, the aSNR of bone, spinal cord and CSF as well as the aCNR of bone/CSF and spinal cord/CSF were calculated as follows:(1)$$aSNR_{tissue}=\frac{{µ}_{tissue}}{\sigma_{tissue}}$$
(2)$$aCNR_{tissue1-tissue2}=\left|\frac{\left({µ}_{tissue1}-{µ}_{tissue2}\right)}{\sqrt{\left(\sigma_{tissue1}{}^{2}+\sigma_{tissue2}{}^{2}\right)}}\right|$$
where *µ* is signal intensity and σ is standard deviation.

### 2.5. Objective Image Analysis: Pixel-Based

In addition to these ROI-based parameters, the pixel-based parameters root mean square error (RMSE) and structural similarity index (SSIM) were calculated. For this purpose, the Digital Imaging and Communications in Medicine (DICOM) images were loaded into an in-house tool that was developed in Python (Python Software Foundation) using the scikit-image toolbox, to perform an automated pixel-wise analysis of the central slice [3,14,24]. The RMSE represents the difference or error of the accelerated sequence compared to the baseline scan (in this case the “standard” 3D sequence), resulting in 0 if the images are identical and higher values for a larger deviation. The RMSE leads to disproportionally large effects if there are differences in signal scaling between compared images. The SSIM provides a percentual deviation for each sequence from the baseline scan with higher values representing greater similarity to the reference image [3,14].

### 2.6. Subjective Image Analysis

Subjective evaluation was independently performed by two board-certified radiologist with 5 years of experience and subspecialization in musculoskeletal imaging (AI, PR). The sequence descriptions were anonymized in our PACS (Picture Archiving and Communication System) and presented to the readers in a random order to avoid any structural effect of consecutive presented scans. Randomization was performed using an online true random integer generator [25]. Delineation and clarity of the following anatomical structures were scored on a 5-point Likert scale: bone marrow, intervertebral disc, spinal cord, CSF, nerve roots and neuroforamina, as well as facet joints (1: not visible/distinguishable, 2: barely visible, 3: adequately visible, 4: good visibility, 5: excellent visibility). Also, the overall image impression was scored on a 5-point Likert scale (1: not acceptable/no diagnostic value, 2: very limited diagnostic value, 3: acceptable for most diagnoses, 4: good for majority of diagnoses, 5: optimal). In addition, the readers were asked to rate ‘yes’ or ‘no’ whether the sequence assessed would be sufficient for clinical use.

### 2.7. Statistical Analysis

Objective and subjective ratings are presented as mean ± SD. Each parameter was tested for normal distribution using the Shapiro–Wilk test. In the case of normal distribution, a repeated measures ANOVA with Geisser–Greenhouse correction and Tukey test for multiple comparisons was performed. In the case of non-parametric without normal distribution, the Friedman test with Dunn’s test for multiple comparisons was performed. A *p*-value of < 0.05 was considered statistically significant. Inter-rater agreement was rated with weighted Cohen’s Kappa (κ). Referring to Landis and Koch [26], the following scale was applied: κ < 0: no agreement, κ between 0.00 and 0.20: slight agreement, κ between 0.21 and 0.40: fair agreement, κ between 0.41 and 0.60: moderate agreement, κ between 0.61 and 0.80: substantial agreement, κ between 0.81 and 1.00: almost perfect agreement [26]. 

Statistical analyses were performed with GraphPad Prism Version 9.2.0 (GraphPad Software Inc., San Diego, CA, USA) and in cases of the calculation of Cohen’s Kappa with the GraphPad QuickCalcs Website (https://www.graphpad.com/quickcalcs/kappa1/ (accessed on 28 November 2022)). [27].

## 3. Results

### 3.1. Study Population

The study population consisted of 7 female and 13 male volunteers with a mean age of 27 ± 7.16 years (range: 20–52 years) and a mean weight of 74.2 ± 12.70 kg (range: 48–92 kg). 

### 3.2. Image Analysis

Examples of the standard-sequence, conventional, and deep-learning-based image reconstructions of undersampled k-data can be found in Figure 2 and Figure 3. The scan duration for the 3D sequences could be reduced with increasing the acceleration factor, as shown in Table 1, along with the further acquisition and reconstruction parameters.

### 3.3. Objective Image Analysis

The results of the objective analysis are summarized in Table 2. For the ROI-based image analysis, aSNR of the bone, spinal cord, and CSF were analysed. When comparing the sequences, a significant main effect could only be demonstrated for aSNR of the bone (*p* = 0.0042), without statistical differences for aSNR of the spinal cord and aSNR of CSF. Further analysis of aSNR of the bone revealed no significant difference when comparing the accelerated sequences with the standard sequence. However, there were statistically significant differences in the comparison of the accelerated sequences with higher aSNR for lower acceleration factors (CS 4.5 vs. CS 11: *p* = 0.0109, CS AI 4.5 vs. CS 11: *p* = 0.0129, CS 8 AI vs. CS 11: *p* = 0.0164) as well as a higher aSNR of the deep-learning-based reconstructions compared to their conventional equivalents (CS 8 vs. CS AI 8: *p* = 0.0030), CS 11 vs. CS AI 11: *p* = 0.0137). Regarding the analysis of aCNR of bone/CSF and spinal cord/CSF, only the aCNR of bone/CSF showed a significant main effect (*p* = 0.0413), however, without significant differences in further comparisons of the sequences.

In the pixel-based comparison, RMSE showed a significant main effect (*p* < 0.0001), as well as significantly lower values when comparing CS 4.5 and CS AI 4.5 with higher acceleration factors (CS 4.5 vs. CS 8: *p* = 0.0002, CS 4.5 vs. CS AI 8, CS 11 and CS AI 11: each *p* < 0.0001, CS AI 4.5 vs. CS 8: *p* = 0.0265, CS AI 4.5 vs. CS AI 8: *p* = 0.0004, CS AI 4.5 vs. CS 11, CS AI 11: each *p* < 0.0001). There were no significant differences between the conventional reconstructions and their deep-learning-based equivalents. Results are depicted in Figure 4. The evaluation of SSIM also showed a significant main effect (*p* < 0.0001), as well as significantly higher values for CS 4.5 and CS AI 4.5 in comparison to sequences of higher acceleration factors (main effect *p* < 0.0001, multiple comparisons: CS 4.5 vs. CS 8: *p* = 0.0002, CS 4.5 vs. CS AI 8: *p* = 0.0006, CS 4.5 vs. CS 11, CS AI 11: each *p* < 0.0001, CS AI 4.5 vs. CS 8: *p* = 0.0042, CS AI 4.5 vs. CS AI 8: *p* = 0.0093, CS AI 4.5 vs. CS 11, CS AI 11: each *p* < 0.00001). Again, there were no significant differences when comparing conventional with deep-learning-based reconstructions. Results are depicted in Figure 5.

### 3.4. Subjective Image Analysis

Interrater agreement was rated with the help of Cohen’s κ, as demonstrated in Table 3, resulting in substantial (κ = 0.61–0.80) or almost perfect (κ = 0.81–1.00) agreement in 94% of cases. Interrater agreement for the use in clinical context yielded a Cohen’s K of 0.743 (substantial agreement). Subjective image analysis is summarized in Table 3.

Further analysis of the results of the subjective reading, as shown in Table 4 and Table 5, revealed significant differences of the sequences regarding the rating of all assessed anatomical structures (bone marrow, intervertebral disc, spinal cord, CSF, nerve roots, and neuroforamina) as well as in the overall image impression (main effect in each case *p* < 0.001). The significance levels of the individual comparisons are listed in Appendix A.

As shown in Table 4, the best ratings were obtained for the sequence CS AI 4.5, which was generally rated better than the standard sequence, with significant differences in the categories “bone marrow”, “intervertebral disc”, and “spinal cord”. The second- and third-best rated sequences were CS 4.5 and CS AI 8, which were rated better than the standard sequence in most cases (except for the category “nerve roots”), although only the comparison of CS 4.5 and standard for the category “bone marrow” reached significance. The other sequences were mostly rated worse than the standard sequence (except for the categories “bone marrow”; intervertebral disc”, and “CSF” in the comparison CS AI 11 vs. standard). The results are depicted using the example of “overall image impression” in Figure 6.

Overall, there was a tendency for better results in the subjective analysis for the AI reconstructions compared with conventional reconstructions at the same acceleration factor, with significant differences when comparing reconstructions with an acceleration factor of 8 (categories “bone marrow”, “spinal cord”, and “CSF”) and 11 (category “bone marrow”). 

In general, there were poorer results at higher acceleration factors. While over 90% of images were classified as sufficient for clinical use at an acceleration factor of 4.5 (CS 4.5: 97.50%, CS AI 4.5: 97.50%), the percentage dropped to 75% at a factor of 8 in the case of conventional reconstructions (CS 8: 75.00%, CS AI 8: 95.00%), and was only 32.50% at an acceleration factor of 11 in the conventional reconstruction (CS 11: 32.50%, CS AI 11: 70.00%).

## 4. Discussion

The purpose of this study was to compare deep-learning-based reconstructions with conventional reconstructions of a Compressed SENSE accelerated 3D T2 sequence of the lumbar spine. In the objective rating, we found a significantly higher aSNR of the bone of the deep-learning-based reconstructions compared to their conventional equivalents for acceleration factors 8 and 11. In the subjective rating, the best results were obtained for CS AI 4.5, CS 4.5, and CS AI 8 sequences. In most cases, these sequences were rated better than the standard sequence, reaching significance in the comparison of the CS AI 4.5 sequence with the standard sequence in three categories and the comparison of the CS 4.5 sequence with the standard sequence in one category. In a direct subjective comparison of deep-learning-based reconstructions with their conventional equivalents, the deep-learning-based reconstructions were rated significantly better in three categories for acceleration factor 8 and in one category for acceleration factor 11.

In summary, the newly developed AI algorithm was non-inferior to the conventional algorithm in all categories and significantly superior in some categories for medium and higher acceleration factors. Translating the subjective results into scan-time reduction, a scan-time reduction to approximately one third is achievable when replacing the standard sequence with the CS AI 8 sequence (149 s vs. 427 s)—a scan time close to that of a conventional 2D T2 sequence (approximately 132 s referring to the clinical standard 2D T2 sequence in our hospital). A substitution of the standard sequence with the CS AI 4.5 sequence already leads to a reduction of the scan time to two thirds (261 s vs. 427 s). Further our data suggest that such substitution increases subjectively perceived image quality at constant objective quality. The corresponding reduction in scan time allows for acquisition of more images per unit time, increases patient comfort, and minimizes the likelihood of motion artifacts. With shorter scanning times, 3D T2 sequences might replace the frequently acquired sagittal 2D sequences of the lumbar spine. High-resolution MRI sequences have proven particularly useful to assess neuroforaminal stenosis of the lumbar spine, which manifest in a parasagittal orientation [28]. In a clinical context, the accurate grading of neuroforaminal stenosis is highly desirable to consistently evaluate the therapeutic concept. Similarly, Foreman et al. found that CS AI high-resolution reconstructions are particularly beneficial for imaging of parasagittally orientated structures, such as the ankle tendons [20]. 

Regarding the direct comparison of the deep-learning-based reconstructions with their conventional equivalents, data of the objective as well as the subjective rating suggest a better image quality of the deep-learning-based reconstructions. Similarly, the only two recent studies using the identical AI-prototype to that used in our research suggest a significant increase of image quality when examining the ankle and prostate compared to standard 3D T2 and CS sequences [20,29]. The quantitative analysis of undersampled ankle and prostate CS AI imagings partially yielded a more than 100% boost of aSNR and aCNR, when compared to standard or CS sequences [20,29]. Within our data, CS AI reconstruction was moderately beneficial for objective image quality, with a maximum of about 10% increase of aSNR and aCNR. In our study, the benefits of AI-driven reconstruction were most obvious in the subjective evaluation of the sequences. While CS AI reconstruction was beneficial in all three available datasets; the objective increase of image quality was more efficient for the anatomy of the ankle and prostate. In addition to different anatomy, another reason for the more extensive effect of CS AI in ankle and prostate imaging might be the resolution of the acquired sequences—previous studies did not assess the objective comparison on high-resolution 3D images but used thicker slices, which is inherently favorable for aSNR and aCNR [20,29].

Our study has several limitations. As we only worked with healthy volunteers, no diagnostic validity of the deep-learning-based reconstructions in terms of pathology detection and assessment can be derived from our study. In addition, we did not work with the default denoising setting but with a strong noise reduction to achieve the best subjective results and to get the best results for the deep-learning-based reconstructions. In return, however, this means that our results may not be fully transferable to other denoising settings. As another minor limitation, this study only evaluated T2-weighted sequences; follow-up studies are needed to assess the possible benefit of this novel technique on further sequences. Following standard practice in recent literature, and since we did not expect major variation of standardized ROI-based measurements of aSNR and aCNR, we refrained from performing an additional inter-reader assessment of the objective analysis.

## 5. Conclusions

The tested deep-learning-based prototype algorithm offers additional potential for scan time reduction in 3D T2 imaging of the lumbar spine using CS AI. It allows for moderate improvement of image quality while significantly reducing scan time compared to the standard SENSE accelerated sequence. Regarding direct comparison of the CS and CS AI approaches, findings of the objective as well as the subjective rating suggest better image quality of the deep-learning-based reconstructions, especially for medium and higher acceleration factors. The development and implementation of deep-learning-based reconstruction algorithms has become more and more important in recent years and might become clinical standard in the future. Therefore, thorough evaluation of their clinical performance needs to be performed for different fields of application. With this study, we provide a first clinical evaluation of a promising prototype that has since been adapted as a clinical product. Future studies might show its applicability in other anatomies and contrasts. Subsequent validation studies are warranted to assess the benefits of this promising reconstruction technology, including clinical and intra-operative correlation.

## Figures and Tables

**Figure 1 diagnostics-13-00418-f001:**
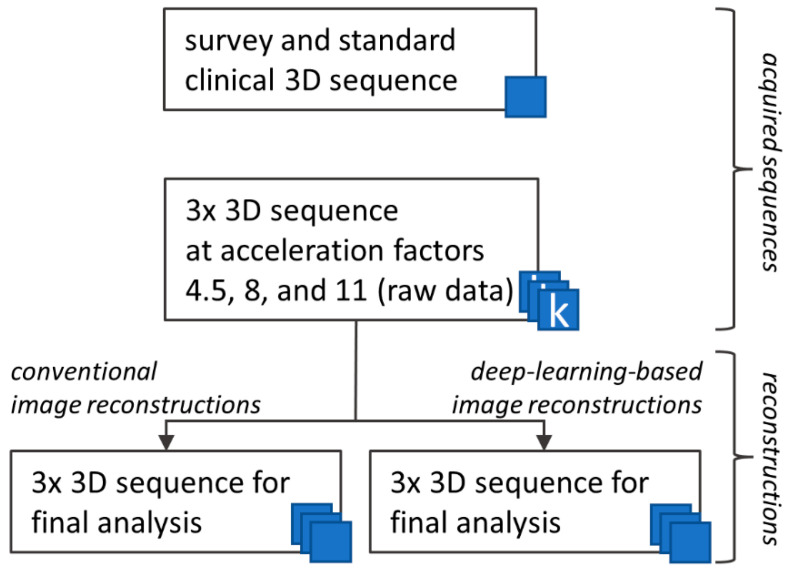
Data acquisition and reconstruction workflow. For each participating individual, we first obtained a survey and standard clinical 3D T2 sequence, which was accelerated by default using parallel imaging (top box, SENSE). Secondly, three sets of 3D k-space raw data were acquired using a combination of parallel imaging and compressed sensing with the acceleration factors 4.5, 8, and 11 (middle box, Compressed SENSE). After that, data acquisition was completed. For subsequent image reconstruction, conventional and deep-learning-based algorithms were used (bottom left and right boxes, respectively). In total, four 3D T2 data sets were acquired, resulting in seven image sets per individual reconstructed for further analysis. Since the conventional and deep-learning images were reconstructed from the identical raw data, a possible bias due to motion artifacts or physiological alterations was precluded from the comparative analysis.

**Figure 2 diagnostics-13-00418-f002:**
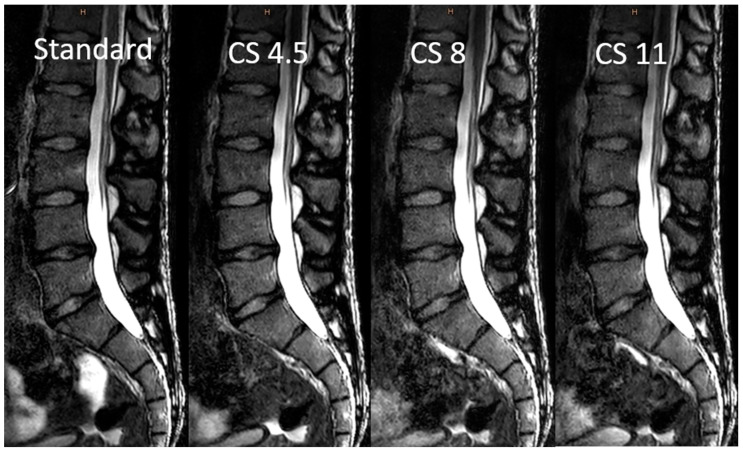
Comparison of the “standard” 3D sequence (without compressed sensing acceleration) and conventional image reconstructions after Compressed SENSE acceleration with factors of 4.5, 8, and 11. Abbreviations: CS, conventional reconstructions of Compressed SENSE images; CS AI, deep-learning-based reconstructions of Compressed SENSE images.

**Figure 3 diagnostics-13-00418-f003:**
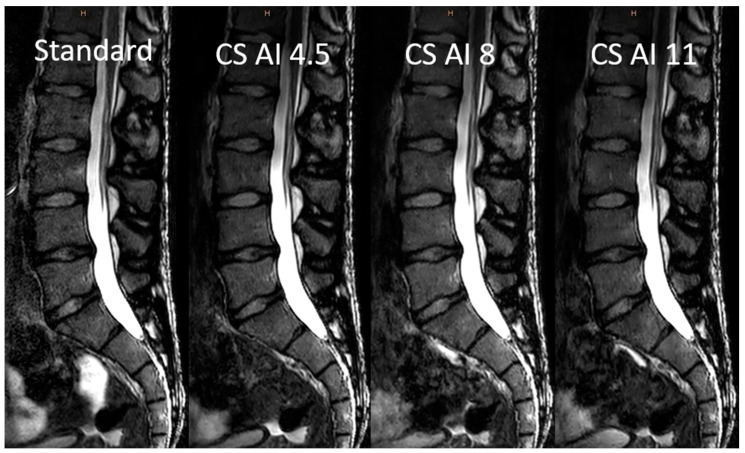
Comparison of the “standard” 3D sequence (without compressed sensing acceleration) and deep-learning-based image reconstructions after Compressed SENSE acceleration with factors of 4.5, 8, and 11. Abbreviations: CS, conventional reconstructions of Compressed SENSE images; CS AI, deep-learning-based reconstructions of Compressed SENSE images.

**Figure 4 diagnostics-13-00418-f004:**
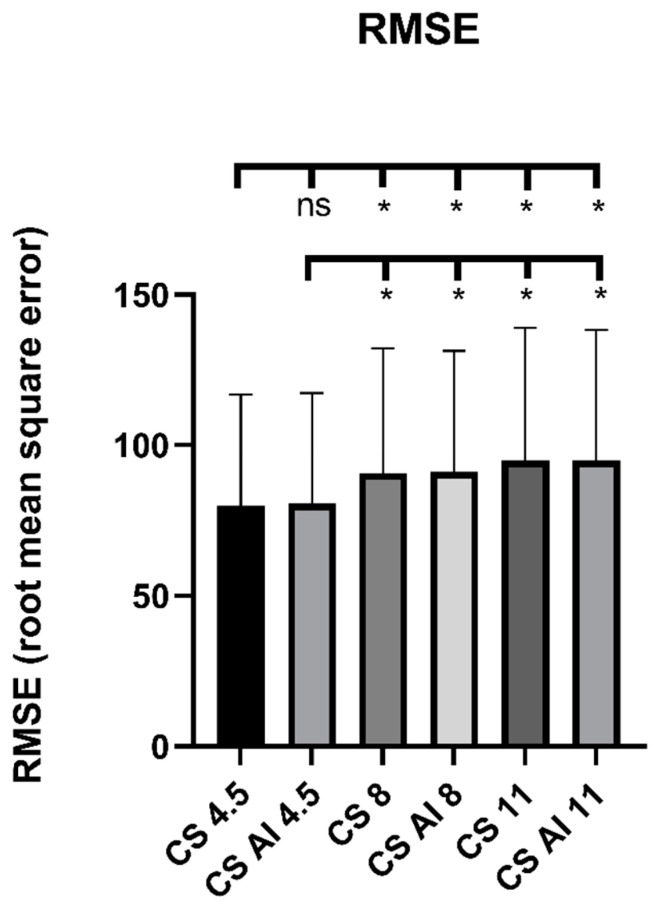
Comparison of sequences with respect to RMSE. Abbreviations: CS = conventional reconstructions of Compressed SENSE images, CS AI = deep-learning-based reconstructions of Compressed SENSE images. *—statistical significance (*p* < 0.05), ns—"not significant" (*p* ≥ 0.05).

**Figure 5 diagnostics-13-00418-f005:**
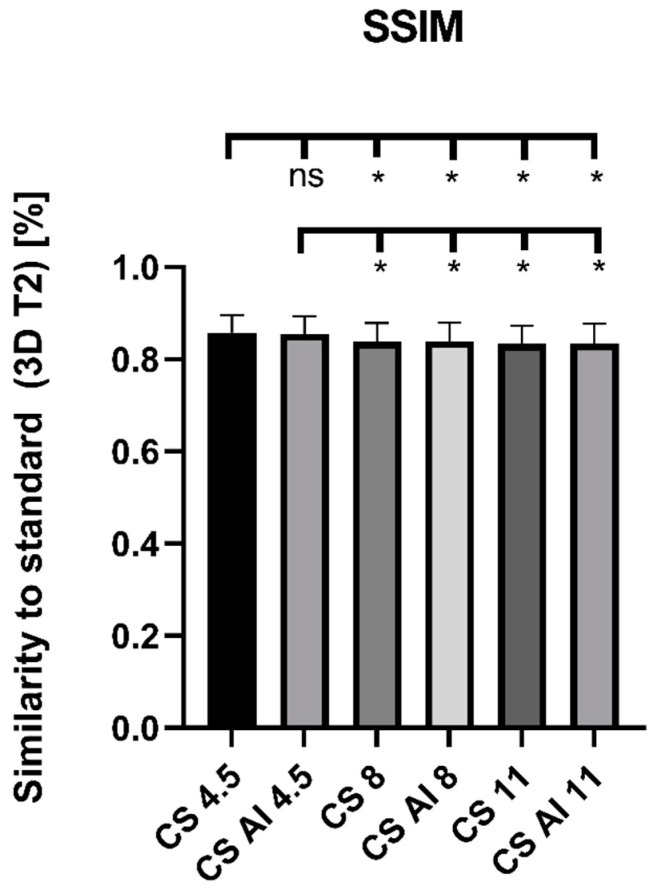
Comparison of sequences with respect to SSIM. Abbreviations: CS = conventional reconstructions of Compressed SENSE images, CS AI = deep-learning-based reconstructions of Compressed SENSE images. *—statistical significance (*p* < 0.05), ns—"not significant" (*p* ≥ 0.05).

**Figure 6 diagnostics-13-00418-f006:**
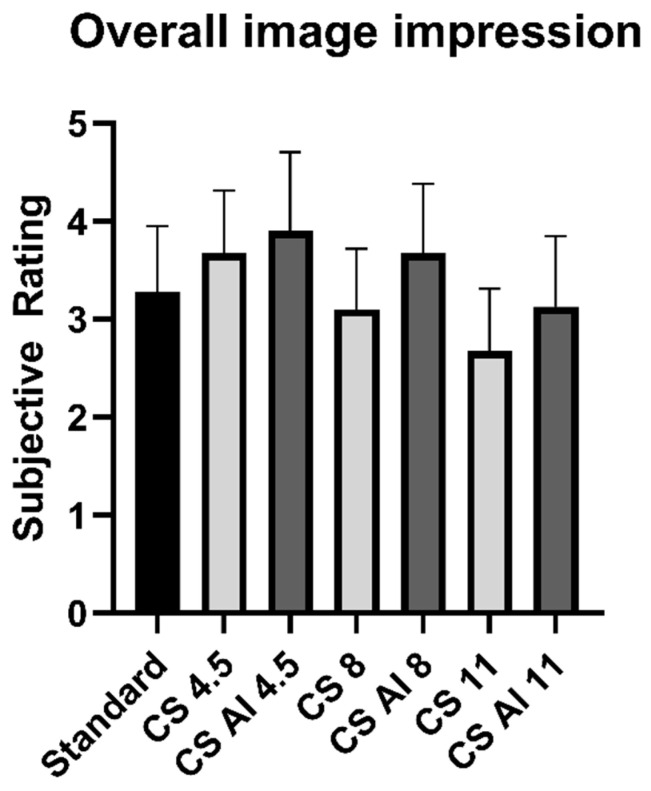
Subjective rating of overall image impression. Abbreviations: CS = conventional reconstructions of Compressed SENSE images, CS AI = deep-learning-based reconstructions of Compressed SENSE images.

**Table 1 diagnostics-13-00418-t001:** Imaging acquisition and reconstruction parameters. Abbreviations: CS, conventional reconstructions of Compressed SENSE images; CS AI, deep-learning-based reconstructions of Compressed SENSE images.

Sequence	Standard 3D	CS 4.5/CS-AI 4.5	CS 8/CS-AI 8	CS 11/CS-AI 11
**Echo time (ms)**	168	168	168	168
**Repetition time (ms)**	3261	3261	3261	3261
**Flip angle (deg.)**	90	90	90	90
**Field of view (mm)**	180 × 300 × 90	180 × 300 × 90	180 × 300 × 90	180 × 300 × 90
**Gap (mm)**	0	0	0	0
**Acquisition voxel size (mm)**	1 × 1 × 1	1 × 1 × 1	1 × 1 × 1	1 × 1 × 1
**Reconstruction voxel size (mm)**	0.47 × 0.47 × 0.5	0.47 × 0.47 × 0.5	0.47 × 0.47 × 0.5	0.47 × 0.47 × 0.5
**Turbo factor/ echo train length**	64	64	64	64
**CS factor**	SENSE 2.5	CS 4.5	CS 8	CS 11
**Scan time (s)**	427	261	149	109
**Saved scan time (s)**	0	166	278	228
**Scan time reduction (%)**	0	38.88	65.11	74.47

**Table 2 diagnostics-13-00418-t002:** Results of objective rating. Abbreviations: CS = conventional reconstructions of Compressed SENSE images, CS AI = deep-learning-based reconstructions of Compressed SENSE images.

	*Standard*	*CS 4.5*	*CS AI 4.5*	*CS 8*	*CS AI 8*	*CS 11*	*CS AI 11*
**RMSE**	-	79.80 ± 37.03	80.53 ± 36.77	90.66 ± 41.37	91.08 ± 40.26	94.87 ± 44.07	94.79 ± 43.49
**SSIM**	-	0.86 ± 0.04	0.86 ± 0.04	0.84 ± 0.04	0.84 ± 0.04	0.83 ± 0.04	0.83 ± 0.04
**aSNR bone**	6.00 ± 1.69	6.51 ± 1.79	6.60 ± 1.97	5.60 ± 1.38	6.18 ± 1.64	5.05 ± 1.21	5.72 ± 1.70
**aSNR spinal cord**	11.36 ± 4.18	10.39 ± 3.65	10.19 ± 3.79	10.56 ± 3.71	10.64 ± 3.61	10.40 ± 3.36	10.22 ± 3.59
**aSNR CSF**	26.26 ± 14.05	28.17 ± 13.53	28.58 ± 13.58	26.89 ± 13.17	27.01 ± 13.06	26.42 ± 12.91	27.69 ± 13.79
**aCNR bone/CSF**	17.41 ± 7.88	19.70 ± 7.70	19.79 ± 7.72	18.46 ± 7.24	18.83 ± 7.25	17.63 ± 6.48	18.57 ± 6.60
**aCNR spinal cord/CSF**	21.72 ± 11.39	23.18 ± 9.90	23.36 ± 9.89	22.26 ± 9.79	22.27 ± 9.79	21.92 ± 9.58	22.63 ± 9.43

**Table 3 diagnostics-13-00418-t003:** Cohen’s Kappa of subjective reading. Abbreviations: CS = conventional reconstructions of Compressed SENSE images, CS AI = deep-learning-based reconstructions of Compressed SENSE images, CSF = cerebrospinal fluid.

	*Standard*	*CS 4.5*	*CS AI 4.5*	*CS 8*	*CS AI 8*	*CS 11*	*CS AI 11*
**Bone marrow**	0.568	0.627	0.743	0.914	0.691	0.733	0.915
**Intervertebral discs**	0.559	0.84	0.916	0.731	0.69	0.821	0.839
**Spinal cord**	0.836	0.655	0.769	0.701	0.846	0.844	0.73
**CSF**	0.914	1.000	0.856	0.774	0.733	0.685	0.688
**Nerve roots**	0.713	0.841	0.918	0.669	0.71	0.491	0.827
**Neuroforamina**	0.931	0.853	0.854	0.712	0.844	0.767	0.838
**Overall image impression**	0.914	0.747	0.923	0.73	0.843	0.754	0.766

**Table 4 diagnostics-13-00418-t004:** Results of subjective reading. Standard refers to the 3D T2 sequence with settings as provided by the manufacturer. Abbreviations: CS = conventional reconstructions of Compressed SENSE images, CS AI = deep-learning-based reconstructions of Compressed SENSE images, CSF = cerebrospinal fluid.

	*Standard*	*CS 4.5*	*CS AI 4.5*	*CS 8*	*CS AI 8*	*CS 11*	*CS AI 11*
**Bone marrow**	3.10 ± 0.81	3.68 ± 0.80	4.03 ± 0.80	2.98 ± 0.66	3.68 ± 0.76	2.45 ± 0.71	3.23 ± 0.73
**Intervertebral discs**	3.00 ± 0.64	3.28 ± 0.68	3.90 ± 0.74	2.78 ± 0.58	3.63 ± 0.81	2.48 ± 0.60	3.10 ± 0.74
**Spinal cord**	3.08 ± 0.73	3.48 ± 0.72	3.85 ± 0.77	2.88 ± 0.82	3.70 ± 0.79	2.83 ± 0.75	3.00 ± 0.68
**CSF**	3.43 ± 0.75	3.70 ± 0.65	3.83 ± 0.87	3.20 ± 0.79	3.78 ± 0.62	3.05 ± 0.75	3.45 ± 0.75
**Nerve roots**	3.72 ± 0.88	3.58 ± 0.75	3.75 ± 0.87	3.38 ± 0.70	3.70 ± 0.85	2.98 ± 0.77	3.28 ± 0.75
**Neuroforamina**	3.45 ± 0.96	3.65 ± 0.86	3.73 ± 0.88	3.05 ± 0.90	3.68 ± 0.76	2.80 ± 0.79	3.28 ± 0.78
**Overall impression**	3.28 ± 0.68	3.68 ± 0.66	3.90 ± 0.81	3.10 ± 0.67	3.68 ± 0.76	2.68 ± 0.69	3.13 ± 0.76

**Table 5 diagnostics-13-00418-t005:** Results of subjective reading regarding the usability of sequences in a clinical context. Abbreviations: CS = conventional reconstructions of Compressed SENSE images, CS AI = deep-learning-based reconstructions of Compressed SENSE images.

	*Standard*	*CS 4.5*	*CS AI 4.5*	*CS 8*	*CS AI 8*	*CS 11*	*CS AI 11*
**Acceptable for use** **in a clinical context**	92.50%	97.50%	97.50%	75.00%	95.00%	32.50%	70.00%

## Data Availability

The data presented in this study are available on request from the corresponding author. The data are not publicly available due to privacy restrictions.

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
