# Peer review of "Conventional and Deep-Learning-Based Image Reconstructions of Undersampled K-Space Data of the Lumbar Spine Using Compressed Sensing in MRI: A Comparative Study on 20 Subjects"

_diagnostics, 2023, doi:10.3390/diagnostics13030418_

Round 1

Reviewer 1 Report (Previous Reviewer 1)

The authors improved the manuscript in comparison to its previous version. I appreciate it may be accepted.

Author Response

Thank you very much for your positive comments!

Reviewer 2 Report (New Reviewer)

The authors aimed to compare deep-learning-based MRI reconstructions with conventional reconstruction of a 3D T2-weighted sequence of the lumbar spine on subjective and objective image quality, which found that compressed sensing in combination with deep-learning-based image reconstruction allows for stronger undersampling of k-space data without loss of image quality, and has potential for further scan time reduction. The current study has been improved after revision; however, a few concerns are needed to be resolved for the current study.

1.     Please consider to add a table to present the MRI protocol, including the main imaging parameters and scan time of each sequence.

2.     The authors used apparent SNR and to CNR to assess objective image analysis, how about the formulators of them? please consider to add them in the manuscript.

3.     Please consider to a table to present the statistical results of subjective image quality, such as sSNR and aCNR.

4.     The authors only assessed and compared deep-learning-based with routine reconstruction methods for 3D T2-weighted images, which is a limitation in this study.

5.     How about the inter-reader agreement of aSNR and aCNR on objective image analysis?

Author Response

Dear Reviewer 2,

Thank you very much for your positive comments on our manuscript. Attached you will find a one-by-one rebuttal considering your recommendations.

  1. Please consider to add a table to present the MRI protocol, including the main imaging parameters and scan time of each sequence.

Please find Table 1 in the final manuscript amended to your recommendation. We added all relevant acquisition and reconstruction parameters, along with the scan time of each sequence.

  1. The authors used apparent SNR and to CNR to assess objective image analysis, how about the formulators of them? please consider to add them in the manuscript.

According to you recommendation, we added the formulas of aSNR and aCNR in a standard format to the respective section, as elaborated below.

(see manuscript=

Formulas 1 and 2. Assessment of apparent signal-to-noise ratio (aSNR) and apparent contrast-to-noise ratio (aCNR).

Abbreviations: µ = signal intensity, σ = standard deviation.

  1. Please consider to a table to present the statistical results of subjective image quality, such as sSNR and aCNR.

As you suggest, the sSNR and aCNR are reported in detail in Table 2, which we highlighted for your convenience.  

  1. The authors only assessed and compared deep-learning-based with routine reconstruction methods for 3D T2-weighted images, which is a limitation in this study.

We added this limitation to the discussion.

  1. How about the inter-reader agreement of aSNR and aCNR on objective image analysis?

Thank you very much for this important comment. Inter-reader analysis is frequently performed to assess reliability of subjective ratings, which we respected in our study. The objective image analysis in our research was performed by an experienced musculoskeletal radiologist in a highly standardized manner, as reported in detail in the Material and Methods section: “ROIs were drawn in the central slice of each sequence in the vertebral body of L1 with an area of 150 mm2, in the spinal cord in segment TH 11/12 with an area of 20 mm2 and in the cerebrospinal fluid (CSF) within the spinal canal in segment L3 with an area of 25 mm2”. Since minimal variation is expected in such measurements, we did not consider an additional inter-reader analysis obligatory. This is also common practice throughout the literature: As a reply to your comment, we performed a PubMed query with the search term “MRI SNR CNR ROI” (20th of January 2023, 11:00 am), and screened the ten most recent peer-reviewed publications for assessment of inter-reader variability of ROI-based SNR and CNR measurements (see below). Among these studies, only An He et al. performed inter-reader variability of standardized ROI-based SNR/CNR measurements and found good agreement between the readers. Hence, following standard practice in recent peer-reviewed literature, we did not add an additional inter-reader analysis of the objective image analysis. To answer your question, we discussed this as a minor limitation in the respective paragraph.

  1. Obama Y, Ohno Y, Yamamoto K, Ikedo M, Yui M, Hanamatsu S, Ueda T, Ikeda H, Murayama K, Toyama H. MR imaging for shoulder diseases: Effect of compressed sensing and deep learning reconstruction on examination time and imaging quality compared with that of parallel imaging. Magn Reson Imaging (2022) 94:56–63. doi:10.1016/J.MRI.2022.08.004
  2. Kohli A, Pilkinton DT, Xi Y, Cho G, Moore D, Mohammadi D, Chhabra A. Image quality improvement and motion degradation reduction in shoulder MR imaging: comparison of BLADE and rectilinear techniques at 3-Tesla scanning. Skeletal Radiol (2022) 51:2291–2297. doi:10.1007/S00256-022-04085-7
  3. Ye Y, Lyu J, Hu Y, Zhang Z, Xu J, Zhang W, Yuan J, Zhou C, Fan W, Zhang X. Augmented T1-weighted steady state magnetic resonance imaging. NMR Biomed (2022) 35:e4729. doi:10.1002/NBM.4729
  4. Zeimpekis KG, Kellenberger CJ, Geiger J. Assessment of lung density in pediatric patients using three-dimensional ultrashort echo-time and four-dimensional zero echo-time sequences. Jpn J Radiol (2022) 40:722–729. doi:10.1007/S11604-022-01258-1
  5. Trudel G, Duchesne-Bélanger S, Thomas J, Melkus G, Cron GO, Larson PEZ, Schweitzer M, Sheikh A, Louati H, Laneuville O. Quantitative analysis of repaired rabbit supraspinatus tendons (± channeling) using magnetic resonance imaging at 7 Tesla. Quant Imaging Med Surg (2021) 11:3460–3471. doi:10.21037/QIMS-20-1343
  6. Zhang X, Wang W, Liu T, Qi Y, Ma L. The effects of three different contrast agents (Gd-BOPTA, Gd-DTPA, and Gd-DOTA) on brachial plexus magnetic resonance imaging. Ann Transl Med (2021) 9:344–344. doi:10.21037/ATM-21-348
  7. An H, Ma X, Pan Z, Guo H, Lee EYP. Qualitative and quantitative comparison of image quality between single-shot echo-planar and interleaved multi-shot echo-planar diffusion-weighted imaging in female pelvis. Eur Radiol (2020) 30:1876–1884. doi:10.1007/S00330-019-06491-3
  8. Jo M, Oh SH. A preliminary attempt to visualize nigrosome 1 in the substantia nigra for Parkinson’s disease at 3T: An efficient susceptibility map-weighted imaging (SMWI) with quantitative susceptibility mapping using deep neural network (QSMnet). Med Phys (2020) 47:1151–1160. doi:10.1002/MP.13999
  9. Lee CH, Vellayappan B, Taupitz M, Hamm B, Asbach P. Dynamic contrast-enhanced MR imaging of the prostate: intraindividual comparison of gadoterate meglumine and gadobutrol. Eur Radiol (2019) 29:6982–6990. doi:10.1007/S00330-019-06321-6
  10. Chassagnon G, Martin C, Ben Hassen W, Freche G, Bennani S, Morel B, Revel MP. High-resolution lung MRI with Ultrashort-TE: 1.5 or 3 Tesla? Magn Reson Imaging (2019) 61:97–103. doi:10.1016/J.MRI.2019.04.015

This manuscript is a resubmission of an earlier submission. The following is a list of the peer review reports and author responses from that submission.

Round 1

Reviewer 1 Report

A positive aspect regarding this study refers to the fact that the data set comprises novel real-world images. However, the number of items is of only 20, hence the significance of the results (including the quality of the statistical tests) is very much reduced. It is not stated how many images are actually obtained from the 20 volunteers. The actual problem that is considered in the study is not adequately introduced; an intuitive presentation of the task should have been included.

The sections and subsections of the article do not flow naturally, they do not appear very connected one to another. Perhaps an author with a general view of the entire study should have made a better cohesion of the entire text. A diagram with a workflow would have helped for a better understanding of the current work.

Reviewer 2 Report

This paper describes the objective and subjective comparison of lumbar spine image reconstruction based on deep learning method and traditional method under different acceleration factors, which proves that the method based on deep-learning is more superior. However, the innovation is not clear and enough, and some formats and content do not meet basic writing requirements. My concerns are as follows:

1. It is a kind of being hard to understand what the proposed idea is useful. As far as I know, the traditional reconstruction of lumbar spine images and the reconstruction of images with deep learning are both existing technologies. Therefore, authors need to emphasize the novelty and advantages of the proposed idea, and provide some more recent literature review in the introduction section.

2. What are the meaning of abbreviations of SENSE, MR, etc., please indicate when they first appear.

3. Section 2, the collected data set is processed through a series of algorithms, so how to judge the accuracy and authenticity of the obtained image data?

4. The article mentioned that it uses deep learning to reconstruct images. Please clearly point out what kind of network was used, and what is the structure of the network?

5. The paper lacks necessary theoretical analysis and derivation, and the readability of the paper decrease.

6. An outlook for future work should be listed in the conclusion.